# RNA-Sequencing Based microRNA Expression Signature of Colorectal Cancer: The Impact of Oncogenic Targets Regulated by *miR-490-3p*

**DOI:** 10.3390/ijms22189876

**Published:** 2021-09-13

**Authors:** Yuto Hozaka, Yoshiaki Kita, Ryutaro Yasudome, Takako Tanaka, Masumi Wada, Tetsuya Idichi, Kan Tanabe, Shunichi Asai, Shogo Moriya, Hiroko Toda, Shinichiro Mori, Hiroshi Kurahara, Takao Ohtsuka, Naohiko Seki

**Affiliations:** 1Department of Digestive Surgery, Breast and Thyroid Surgery, Graduate School of Medical and Dental Sciences, Kagoshima University, Kagoshima 890-8520, Japan; k6958371@kadai.jp (Y.H.); north-y@m.kufm.kagoshima-u.ac.jp (Y.K.); k7682205@kadai.jp (R.Y.); k5495007@kadai.jp (T.T.); k8911571@kadai.jp (M.W.); k3352693@kadai.jp (T.I.); k3113670@kadai.jp (K.T.); k4900207@kadai.jp (H.T.); morishin@m3.kufm.kagoshima-u.ac.jp (S.M.); h-krhr@m3.kufm.kagoshima-u.ac.jp (H.K.); takao-o@kufm.kagoshima-u.ac.jp (T.O.); 2Department of Functional Genomics, Graduate School of Medicine, Chiba University, Chiba 260-8670, Japan; cada5015@chiba-u.jp; 3Department of Biochemistry and Genetics, Graduate School of Medicine, Chiba University, Chiba 260-8670, Japan; moriya.shogo@chiba-u.jp

**Keywords:** microRNA, colorectal cancer, RNA-sequencing, expression signature, tumor-suppressor, *miR-490-3p*, *IRAK1*

## Abstract

To elucidate novel aspects of the molecular pathogenesis of colorectal cancer (CRC), we have created a new microRNA (miRNA) expression signature based on RNA-sequencing. Analysis of the signature showed that 84 miRNAs were upregulated, and 70 were downregulated in CRC tissues. Interestingly, our signature indicated that both guide and passenger strands of some miRNAs were significantly dysregulated in CRC tissues. These findings support our earlier data demonstrating the involvement of miRNA passenger strands in cancer pathogenesis. Our study focused on downregulated *miR-490-3p* and investigated its tumor-suppressive function in CRC cells. We successfully identified a total of 38 putative oncogenic targets regulated by *miR-490-3p* in CRC cells. Among these targets, the expression of three genes (*IRAK1*: *p* = 0.0427, *FUT1*: *p* = 0.0468, and *GPRIN2*: *p* = 0.0080) significantly predicted 5-year overall survival of CRC patients. Moreover, we analyzed the direct regulation of *IRAK1* by *miR-490-3p*, and its resultant oncogenic function in CRC cells. Thus, we have clarified a part of the molecular pathway of CRC based on the action of tumor-suppressive *miR-490-3p*. This new miRNA expression signature of CRC will be a useful tool for elucidating new molecular pathogenesis in this disease.

## 1. Introduction

Colorectal cancer (CRC) is one of the most common cancers of the digestive system, as it constitutes approximately 10% of all diagnosed cancers, with almost 900,000 deaths annually [1,2]. The expansion of treatment options and the spread of diagnostic systems have greatly contributed to improving the prognosis of patients with CRC [2,3,4]. However, for patients with advanced cancer with metastasis, the 5-year survival rate is approximately 14%, and improving the prognosis of these patients is an important clinical issue [3]. Furthermore, the number of young people (under the age of 50) with colorectal cancer is increasing, especially rectal cancer and left-sided colon cancer [2].

Over the past few decades, the mechanism of CRC oncogenesis has been revealed by molecular genetic analyses [5,6,7]. A previous study demonstrated that several genes are frequently mutated, e.g., *APC*, *TP53*, *SMAD4*, *KRAS*, *PIK3CA*, *ARID1A*, *SOX9,* and *FAM123B*, and pivotal oncogenic pathways, e.g., WNT, RAS-MAPK, PI3K, TGF-β, P53, and DNA mismatch-repair are closely associated with this disease [7]. Notably, for younger CRC patients, the mutation rate of these genes is low, and it is expected that there are other factors involved [8].

The Human Genome Project revealed that a large number of non-protein-coding RNA molecules (ncRNAs) are transcribed from the human genome [9]. Current studies indicate that a large number of ncRNAs play critical roles in various biological processes, e.g., stabilization of RNA molecules, regulation of gene expression, and control of the cell cycle [10,11]. Notably, it is evident that dysregulation of ncRNA is involved in the enhancement of human diseases, including cancers [12].

Functional ncRNAs are classified by their nucleotide length: short ncRNAs (19–30 nucleotides), medium ncRNAs (50–200 nucleotides), and long ncRNAs (>200 nucleotides) [13]. MicroRNAs (miRNAs) constitute the most commonly analyzed class of short ncRNAs. They function as fine-tuners of gene expression, working in a sequence-dependent manner [14]. One of the hallmarks of miRNAs is that a single miRNA regulates multiple RNA transcripts. Moreover, they contribute to various cellular signaling pathways in both physiological and pathological conditions [14,15]. A vast number of studies have shown that ectopic expression of miRNAs modulates oncogenes and/or tumor-suppressors in human cancer cells, including CRC [16,17,18].

We have identified tumor-suppressive miRNAs, and their regulated oncogenes, in several types of cancers [19,20,21]. Our previous studies revealed that the expression of tumor-suppressive miRNAs and regulated oncogenes are closely related to the molecular pathogenesis of cancers. Our miRNA-based strategy is an attractive approach to the identification of novel oncogenic genes/pathways in cancer cells.

In this study, we attempted to identify novel dysregulated miRNAs in CRC tissues. Thus, we created a new miRNA expression signature using next-generation sequencing technology. Currently available RNA sequencing technologies are suitable for rapidly, and accurately, generating miRNA signatures [22]. Our new CRC signature showed that a total of 154 miRNAs (84 upregulated and 70 downregulated) were significantly dysregulated in CRC tissues. Here, we focused on *miR-490-3p* (the most downregulated miRNA in this signature), and we investigated its tumor-suppressive roles and its targeted oncogenes in CRC cells. Finally, we identified *IRAK1* as an oncogenic *miR-490-3p* target and demonstrated that its expression is closely involved in CRC molecular pathogenesis.

Using tumor-suppressive *miR-490-3p* as a starting point, we have succeeded in identifying molecules involved in the oncogenesis of CRC. The CRC miRNA signature we provide in this study will contribute to the search for regulatory molecular networks in CRC.

## 2. Results

### 2.1. Creating miRNA Expression Signature in CRC

A total of 10 samples (5 CRC tissues and 5 noncancerous tumor-adjacent tissues) were analyzed using RNA sequencing techniques to generate a CRC miRNA signature. The characteristics of five patients with CRC are summarized in Table 1. In this analysis, we obtained between 11,432,618 and 19,286,983 total sequence reads (Appendix A). After the trimming procedure, we mapped them to the human genome and determined that 3,018,259 to 16,436,280 sequences were small human RNAs (Appendix A). Our RNA sequence data was deposited in the Gene Expression Omnibus (GEO) database (GSE183437).

Differentially expressed miRNAs were mapped on human chromosomes (Figure 1 and Figure 2). Upregulated and downregulated miRNAs (84 and 70, respectively) were aberrantly dysregulated in CRC tissues (Appendix A). Interestingly, among the abnormally downregulated miRNAs in CRC tissue, 16 pairs of miRNAs (i.e., *miR-490*, *miR-133a*, *miR-145*, *miR-129*, *miR-143*, *miR-497*, *miR-9*, *miR-139, miR-125b*, *miR-100*, *miR-30a*, *miR-218*, *miR-195*, *miR-99a*, *miR-29c*, and *miR-28*) represented both *miR-5p and -3p* sequences. 

### 2.2. Tumor-Suppressive Roles of miR-490-5p and miR-490-3p in CRC Cells

The *miR-490-*duplex was the most downregulated in this miRNA expression signature (Figure 3A). In this study, we investigated the functional and oncogenic significance of both strands of the *miR-490-*duplex in CRC cells. To confirm the validity of the CRC signature, we measured the expression levels of *miR-490-3p* and *miR-490-5p* in clinical specimens (27 CRC specimens and 27 tumor-adjacent normal colorectal epithelial specimens). The characteristics of 27 patients with CRC are summarized in Appendix A. Expression levels of both *miR-490-3p* and *miR-490-5p* were significantly reduced (*p* < 0.0001) in CRC tissues compared with those in normal colorectal epithelial tissues, assessed by quantitative polymerase chain reaction (qPCR). (Figure 3B). The expression levels of these miRNAs in two cell lines (HCT116 and DLD-1) were lower than those in normal colorectal epithelial tissues (Figure 3B). There was a positive correlation between the expression levels of the two miRNAs by Spearman’s rank analysis (*r* = 0.957, *p* < 0.0001; Figure 3C).

To investigate the antitumor functions of *miR-490-3p* and *miR-490-5p* in CRC cells, we performed ectopic expression assays. The experiments were performed by a transient transfection method using mature miRNAs, *miR-490-3p* (caaccuggaggacuccaugcug), or *miR-490-5p* (ccauggaucuccaggugggu). Cell proliferation, migration and invasive ability were significantly attenuated by *miR-490-3p* transfection into CRC cell lines (Figure 4A–C). In contrast, cell migration abilities were not blocked by *miR-490-5p* transfection into CRC cell lines (Figure 4C).

### 2.3. MiR-490-3p and miR-490-5p Was Incorporated into the RNA-Induced Silencing Complex (RISC) in CRC Cells

In miRNA biogenesis, it is essential that miRNAs are incorporated into the RISC to regulate target genes. Ago2 is a fundamental component of the RISC. We investigated that *miR-490-3p* and *miR-490-5p* actually functioned in CRC cells by immunoprecipitation using an anti-Ago2 antibody. Levels of the miRNA incorporation into Ago2 were quantified with qPCR. Mature *miR-490-3p* was transfected into CRC cells, and it was incorporated into the RISC. Levels of *miR-490-3p* were significantly higher than that in cells transfected with mock, miR-control, and *miR-490-5p* (Appendix A). Similarly, when transfected with mature *miR-490-5p*, the levels of *miR-490-5p* were significantly higher than those of cells transfected with mock, miR-control, and *miR-490-3p* (Appendix A). These results indicated that *miR-490-3p* and *miR-490-5p* were incorporated into RISC separately.

### 2.4. Screening of Putative Oncogenic Targets by miR-490-3p Regulation in CRC Cells

Our screening strategy of *miR-490-3p* targets is shown in Figure 5. To identify the putative oncogenic targets of *miR-490-3p* regulation in CRC cells, we assessed three datasets, i.e., TargetScan database, the GEO database (genes downregulated in *miR-490-3p*-transfected HCT116), and TCGA-COAD database through the GEPIA2 platform (genes upregulated in CRC tissues).

There were 38 candidate targets for *miR-490-3p* regulation in CRC cells (Table 2), of which three genes (*IRAK1*, *FUT1*, and *GPRIN2*) were associated with poor prognosis (Figure 6). Among these targets, we focused on *IRAK1* as an oncogenic target of *miR-490-3p* regulation in CRC cells.

### 2.5. Direct Regulation of IRAK1 by miR-490-3p in CRC Cells

We investigated direct regulation of *IRAK1* by *miR-490-3p* in CRC cells. Expression levels of *IRAK1* mRNA and IRAK1 protein were significantly reduced by aberrant expression of *miR-490-3p* in CRC cells (HCT116 and DLD-1, Figure 7A,B).

Next, a dual luciferase reporter assay was performed to assess whether *miR-490-3p* bound directly to the *IRAK1* target site. Luciferase activity was significantly reduced after co-transfection of *miR-490-3p* and a vector carrying the wild-type *miR-490-3p* target site. In contrast, luciferase activity was not changed following co-transfection of *miR-490-3p* and a vector carrying the deletion-type *miR-490-3p* target site (Figure 7C). These results showed that *IRAK1* was directly regulated by *miR-490-3p* in CRC cells.

### 2.6. Effects of IRAK1 Knockdown in CRC Cells

To investigate the expression of *IRAK1* mRNA/IRAK1 protein in CRC cells, we performed CRC knockdown assays using two different siRNAs. The expression levels of both *IRAK1* mRNA and *IRAK1* protein were markedly reduced by *siIRAK1*-1 and *siIRAK1*-2 in the two cell lines (Figure 8A,B).

By suppressing *IRAK1* expression, cell proliferation assays showed no significant effects of these siRNA transfected into the two cell lines (Figure 9A), but migration and invasive abilities were significantly blocked in CRC cells (Figure 9B,C). These data suggested that aberrant expression of *IRAK1* promoted cancer-related phenotypes in CRC cells.

### 2.7. Expression of IRAK1 in CRC Clinical Specimens

We analyzed expression levels of *IRAK1* (RNA-Sequence data in 275 colon cancer tissues compared to 349 normal colon tissues) using TCGA-COAD database through the GEPIA2 platform. Expression of *IRAK1* was significantly upregulated (*p* < 0.01; Figure 6). In addition, immunohistochemistry was used to assess protein expression levels of *IRAK1* in CRC clinical specimens, and high expression of *IRAK1* were shown in cancer lesions (Figure 10).

## 3. Discussion

Clarifying the molecular pathogenesis of CRC based on the latest genomic analyses will contribute to the development of new treatment strategies for this disease. We have elucidated novel molecular pathogenesis using a miRNA-based approach in various carcinomas [26,27]. To date, high-throughput technologies (e.g., oligo-microarrays, PCR-based arrays, and RNA sequences) have enabled the construction of CRC miRNA expression signatures, revealing aberrant expression of many miRNAs [24,28,29,30,31,32,33,34]. Previous studies have shown that *miR-490-3p*, *miR-195-5p,* and *miR-30a-5p*, which are frequently downregulated in CRC, function as tumor-suppressive miRNA in CRC cells [25,35,36,37,38,39]. These miRNAs were included in the signature we created in this study. Importantly, for 16 pairs downregulated, both miRNA strands (−5p and −3p) were ectopically expressed in cancer tissues. Recently, it was suggested that the initial hypothesis that one of the two strands is degraded during miRNA biosynthesis may be incorrect [40]. Our recent studies show that some passenger miRNAs have a tumor suppressor function in cancer cells. (e.g., *miR-148-5p*, *miR-145-3p*, *miR-143-5p*, *miR-30c-2-3p*, and *miR-30a-3p*) [19,26,41,42]. Those miRNA duplexes and their target oncogenic genes are closely associated with cancer pathogenesis [19,26,41,42].

Growing body of studies showed that downregulation of *miR-490-5p* and *miR-490-3p* were closely associated with a wide range of human cancers [43]. Among them, we focused on *miR-490-3p*, which was the most down-regulated in our newly created signature. Previous study showed that downregulation of *miR-490-3p* was reported by high-throughput sequence-based analysis in CRC [24,25]. Moreover, several studies have shown that downregulation of *miR-490-3p* occurs frequently in several types of cancer (e.g., breast cancer, lung adenocarcinoma) and that this miRNA acts as a tumor suppressor [44,45,46]. These reports indicate that downregulation of *miR-490-3p* has a critical effect on human tumorigenesis, and it is an important issue to identify for target molecules of *miR-490-3p* in each cancer type.

Previous studies showed that tumor-suppressive function of *miR-490-3p* in CRC cells through targeting several oncogenic genes [35,36,47,48]. For example, ectopic expression of *miR-490-3p* blocked migration and invasion abilities in CRC cells [47]. Transfection of *miR-490-3p* inhibited cancer cell malignant phenotypes, e.g., cells proliferation, metastasis, invasion, and anti-apoptosis [36]. In addition, *miR-490-3p* directly regulated *VDAC1* expression, and a negatively controlled VDAC1/AMPK/mTOR pathway [36]. Ectopic expression of *miR-490-3p* attenuated to cancer cell malignant transformation both in vitro and in vivo [48]. Additionally, *miR-490-3p* directly controlled *RAB14* in CRC cells [48]. Expression of *miR-490-3p* affected cell viability and resistance to chemotherapy in CRC cells through regulating *TNKS2* [35]. Our data also showed that ectopic expression of *miR-490-3p* significantly blocked malignant phenotypes of CRC cells, which is completely consistent with previous reports.

Next, we aimed to elucidate *miR-490-3p*-regulated oncogenes and oncogenic pathways in CRC cells. Our in silico analysis revealed that three genes (*IRAK1*, *FUT1*, and *GPRIN2*) were associated with poor prognosis. In this study, we focused on *IRAK1* (interleukin-1 receptor-related kinase 1) because this kinase family is a target of drug discovery.

*IRAK1* is one of the serine-threonine kinases that mediates the signaling pathways of toll-like receptors and the inflammatory mediator interleukin-1 [49,50]. Recent studies have revealed that *IRAK1* is involved not only in inflammatory diseases but also in progression of several cancers [51,52,53,54]. In vivo, inhibition of *IRAK1*, in mice with colitis-induced tumorigenesis, reduced the inflammatory response and inhibited the epithelial–mesenchymal transition [55]. In the current study, we focused on *IRAK1* and showed that its aberrant expression was closely associated with CRC malignant phenotypes. Further functional analyses of *IRAK1* will possibly reveal the biological characteristics of CRC. Starting from tumor-suppressive miRNA, we identified effective prognostic markers and therapeutic targets for CRC, indicating that our miRNA-based strategy was feasible.

## 4. Materials and Methods

### 4.1. Patient Samples

CRC tissues and noncancerous tumor-adjacent tissues (27 each) were used to verify the expression status of *miR-490-5p* and *miR-490-3p* (Appendix A). All samples were collected from patients who underwent surgical resection at Kagoshima University Hospital between 2014 and 2017. Written informed consent for the use of their specimens was obtained from all patients. The study was conducted according to the guidelines of the Declaration of Helsinki and approved by Ethics Committee of Kagoshima University (approval no. 160,038 28-65, date of approval: 19 March 2021)

### 4.2. CRC Cell Lines and Cell Culture

The two human CRC cell lines (HCT116 and DLD-1) were used in this study. HCT116 cells were obtained from the RIKEN Cell Bank (Tsukuba, Ibaraki, Japan) and DLD-1 cells were obtained from the Cell Resource Center for Biomedical Research/Cell Bank (Sendai, Miyagi, Japan). HCT116 was maintained in DMEM supplemented with 10% fetal bovine serum (FBS). DLD-1 was cultured in RPMI-1640 medium with 10% FBS.

### 4.3. Small RNA Sequencing

The small RNA sequencing and data mining process were performed as in previous studies [19,21]. Briefly, flesh frozen on dry ice using 10 samples (5 CRC tissues, and RNA was harvested using Trizol reagent). Illumina TruSeq Small RNA Sample Preparation Kit was used for the construction of sequencing libraries. miRNA libraries were prepared for sequencing using standard Illumina protocols. New miRNA expression signatures were generated using a next-generation sequencer HiSeq 2500 (Illumina, San Diego, CA, USA). Sequenced reads were trimmed for adaptor sequence, and masked for low-quality sequence using cutadapt v.1.2.1. A false discovery rate (FDR) less than 0.05 was considered significant. The present RNA sequencing data was deposited in GEO database (GSE183437).

### 4.4. RNA Extraction and qPCR

We performed RNA extraction from clinical samples, cell lines, and qRT-PCR using the methods we described previously [16,21]. According to the manufacturer’s protocol, total RNA was isolated from flesh frozen colorectal tissues and cell lines using TRIzol reagent. RNA quality was confirmed using an Agilent 2100 Bioanalyzer (Agilent Technologies, Santa Clara, CA, USA). Then, RNA sample reverse transcription was achieved with a High Capacity cDNA Reverse Transcription Kit (Applied Biosystems, Waltham, MA, USA). The qPCR was performed with a PCR Master Mix (Applied Biosystems, Waltham, MA, USA) real-time PC detection system (BioRad Laboratories, Hercules, CA, USA). *RNU48* and *GUSB* were used as normalized controls. The reagents used in the analysis are listed in Appendix A.

### 4.5. Transfection of Mature miRNAs, Small-Interfering RNAs, and Plasmid Vectors into CRC Cells

The miRNA precursors and siRNAs were obtained by Invitrogen (Thermo Fisher Scientific, Waltham, MA, USA). Transfection of miRNA precursors, siRNAs, or negative control miRNA/siRNA was performed with Lipofectamine™ RNAiMAX and that of plasmid vectors was performed with Lipofectamine™ 2000 (Thermo Fisher Scientific, Waltham, MA, USA). HCT116 and DLD-1 cells were transfected with 10 nM miRNA, siRNA, or negative control miRNA/siRNA. The reagents used in the process are listed in Appendix A.

### 4.6. Cell Proliferation, Migration and Invasion Assays in CRC Cells

The methods used for functional assessment of CRC cells (e.g., proliferation, invasion, and migration) were outlined in previous studies [16,21]. In brief, for proliferation assays, cells were transferred to 96-well plates. HCT116 or DLD-1 cells were plated at 1.0 × 10^4^ cells per well. After 72 h, cell proliferation was evaluated using XTT assays. For migration and invasion assays, HCT116 cells or DLD-1 cells at 1.2 × 10^5^ were transfected in 6-well plates. After 72 h, HCT116 cells or DLD-1 cells were added into each chamber at 2.5 × 10^5^ per well. After 48 h, the cells on the lower surface were counted for analysis. All experiments were performed in triplicate.

### 4.7. Assay of miR-490-3p Incorporation into the RNA-Induced Silencing Complex (RISC)

Mature *miR-490-3p* and *miR-490-5p* were separately transfected into 1.2 × 10^5^ CRC cells (HCH116 and DLD-1) per ml. After 72 h, miRNA incorporated into the RISC were isolated using a human AGO2 miRNA isolation kit (Wako Pure Chemical Industries, Ltd., Osaka, Japan) according to the manufacturer’s protocol. Amount of incorporated miRNA was evaluated by RT-qPCR as described previously [56]. *miR-21* was used as the internal control.

### 4.8. Candidate Target Genes Controlled by miR-490-3p in CRC Cells

To identify genes targeted by *miR-490-3p*, we obtained microarray data for HCT116 cells transfected with *miR-490-3p*. By combining these data with the Target Scan Human 7.2 database (http://www.targetscan.org/vert_71 (accessed on 12 May 2021)), we extracted a total of 249 possible target genes with *mi-490-3p* binding sites. Next, we analyzed the gene expression levels of 275 colon cancer tissues and 349 normal colon tissue samples in the Cancer Genome Atlas (TCGA)-COAD database via the GEPIA2 platform (https://cancergenome.nih.gov/ (accessed on 2 May 2021)) and thereby identified 38 oncogenic genes. Among those 38 genes, the expression of 3 genes showed statistically significant correlations with the 5-year overall survival rates of patients with CRC obtained from OncoLnc (http://www.oncolnc.org/ (accessed on 9 May 2021)). Our microarray data were deposited in the GEO database (GSE129043).

### 4.9. Dual-Luciferase Reporter Assays

The predicted binding site sequence of *miR-490-3p* in the 3′-UTR of *IRAK1* was extracted from the TargetScanHuman database (https://www.targetscan.org/, release 7.2 (accessed on 12 May 2021)). Based on those data, a PsiCHECK-2 plasmid vector (Promega, Madison, Wisconsin, USA) containing wild-type was used. To generate the *IRAK1* mutant reporter, the seed region of the *IRAK1* 3′-UTR was mutated to remove all complementarity to nucleotides of *miR-490-3p*. CRC cells were seeded into a 24-well plate. After being cultured overnight, CRC cells were co-transfected with the indicated vectors and miR precursor of *miR-490-3p* or negative control miRNA. Luciferase assays were performed 60 h after transfection using the Dual Luciferase Reporter Assay System (Promega). Normalized data were calculated as the Renilla/firefly luciferase activity ratio.

### 4.10. Western Blotting and Immunohistochemistry

The procedures for Western blotting and immunostaining were described previously [16,21]. Briefly, 21 µg of protein lysates were separated on 4–20% SDS PAGE Gel and transferred to PVDF membranes (Thermo Fisher Scientific). Membranes were blocked with skim milk and incubated with the indicated primary antibodies overnight at 4 °C. The antibodies used in this study are shown in Appendix A. GAPDH was used as the internal control. We assessed expression of IRAK1 proteins by immunohistochemistry. The procedure for immunostaining was described previously.

### 4.11. Statistical Analyses

Differences between 2 groups were evaluated using Mann–Whitney U tests. Correlation coefficients were evaluated using Spearman’s test. All statistical analyses were performed using JMP Pro 15 (SAS Institute Inc., Cary, NC, USA). *p*-values < 0.05 were considered statistically significant, and all data are presented as the mean ± standard error (SE).

## 5. Conclusions

We created new RNA sequencing-based CRC miRNA signatures that revealed novel miRNAs that were aberrantly expressed. This miRNA signature laid the foundation for exploring a new molecular RNA network for CRC. Moreover, we revealed that *IRAK1*, regulated by *miR-490-3p*, may be a novel diagnostic and therapeutic target in CRC. Our approach, which utilized the analysis of miRNA signatures, could contribute to the elucidation of the molecular pathogenesis of cancer.

## Figures and Tables

**Figure 1 ijms-22-09876-f001:**
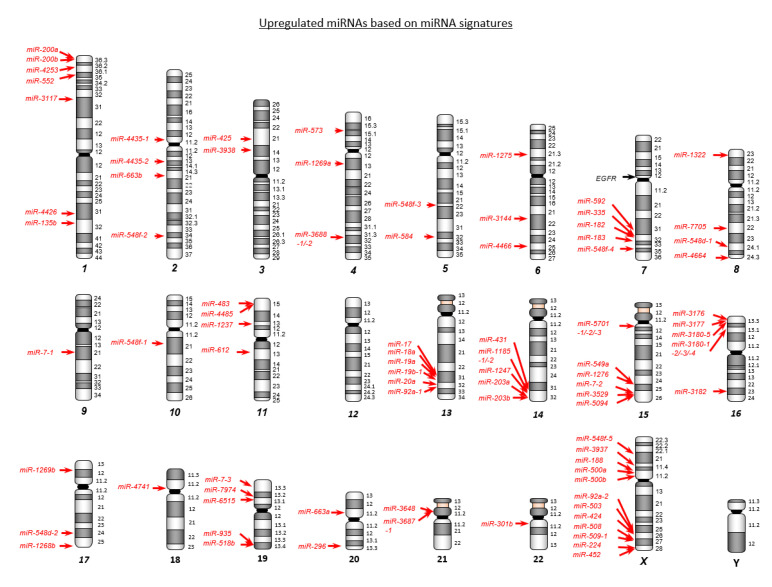
Chromosome mapping of aberrantly upregulated miRNAs in colorectal cancer (CRC). Upregulated miRNAs in CRC tissues are mapped on human chromosomes. A total of 84 miRNAs are identified by our RNA sequence based signature.

**Figure 2 ijms-22-09876-f002:**
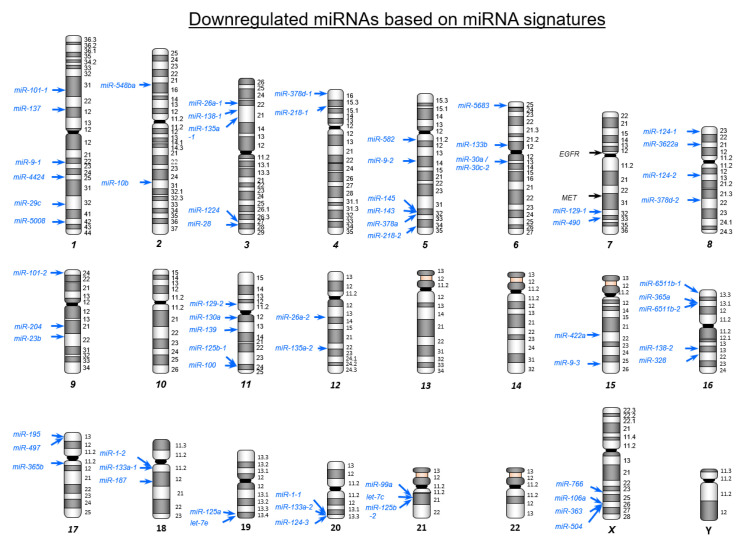
Chromosome mapping of aberrantly downregulated miRNAs in colorectal cancer (CRC). Downregulated miRNAs in CRC tissues are mapped on human chromosomes. A total of 70 miRNAs are identified by our RNA sequence based signature. In order to ensure the validity of our signature created in this study, we compared to previous miRNA signatures of CRC [23,24,25]. Two miRNAs, *miR-592* (upregulated miRNA) and *miR-139-5p* (downregulated miRNA) were common miRNAs to all four independent signatures (Appendix A). Eight miRNAs, *miR-335-3p*, *miR-552-5p*, *miR-3180-5p*, *miR-301b* and *miR-3144-3p* (upregulated miRNA), and *miR-378a-5p*, *miR-490-3p,* and *miR-422a* were common to multiple independent signatures (Appendix A). These signatures suggest that aberrant expressed miRNAs, listed in multiple studies, play a vital role in CRC tumorigenesis.

**Figure 3 ijms-22-09876-f003:**
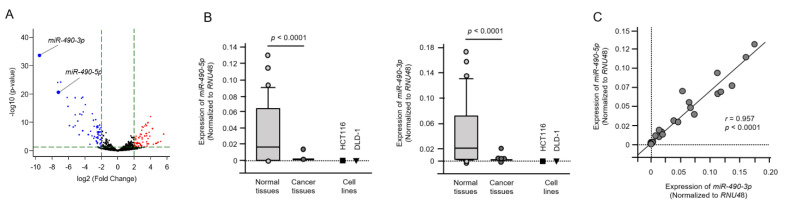
Expression of *miR-490-5p* and *miR-490-3p* by small RNA sequencing and expression of *miR-490-5p* and *miR-490-3p* in colorectal cancer (CRC) tissues and cell lines. (**A**) Volcano plot of the miRNA expression signature determined by RNA sequencing. The log2 fold-change (FC) is plotted on the *x*-axis, and the −log10 (*p*-value) is plotted on the *y*-axis. The blue points represent the downregulated miRNAs with an absolute −log2 FC ≥ 1 (FC = 2) and *p*-value < 0.05. (**B**) The expression levels of *miR-490-5p* and *miR-490-3p*, by quantitative polymerase chain reaction, were evaluated in CRC clinical tissues and cell lines (HCT116 and DLD-1). Expression levels of these miRNAs were significantly reduced in cancer tissues (*p* < 0.001). (**C**) Spearman’s rank test showed positive correlations between the expression levels of *miR-490-5p* and *miR-490-3p* in clinical specimens (*r* = 0.957, *p* < 0.001).

**Figure 4 ijms-22-09876-f004:**
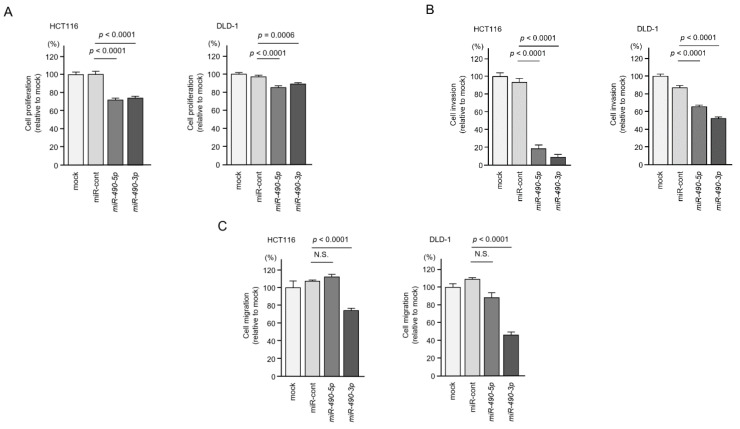
Tumor-suppressive roles of *miR-490-5p* and *miR-490-3p* in colorectal cancer (CRC) cells (HCT116 and DLD-1). Functional assays of *miR-490-5p* and *miR-490-3p* in CRC cells using miRNAs precursors. (**A**) Cell proliferation, assessed by XTT assay, 72 h after transfection of mature miRNAs (**B**) Cell invasion, determined by Matrigel invasion assay, 48 h after seeding inhibitor-transfected cells into the chambers. (**C**) Cell migration assessed, using a membrane culture system, 48 h after seeding inhibitor-transfected cells into the chambers.

**Figure 5 ijms-22-09876-f005:**
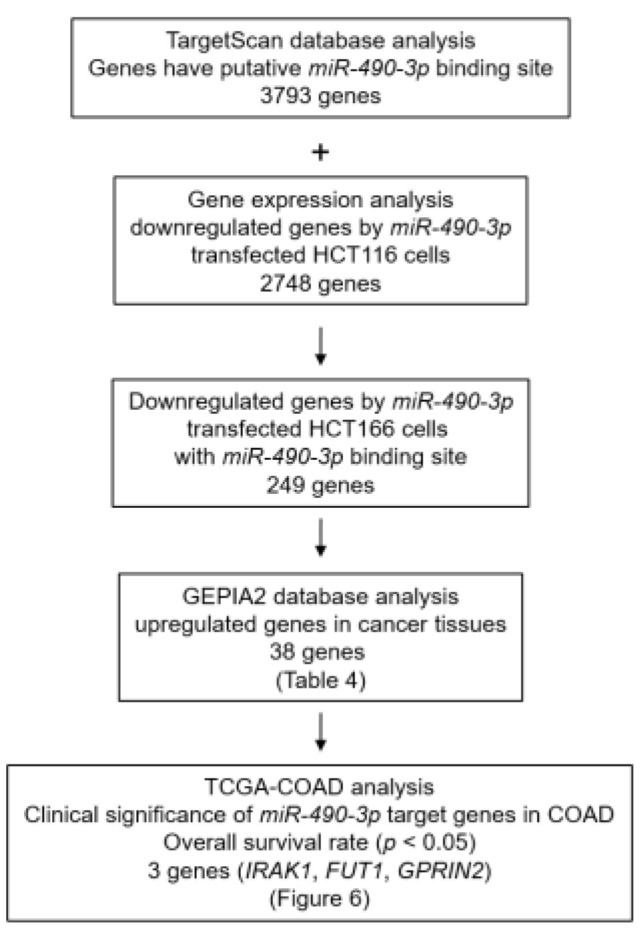
Flowchart for searching for oncogenic targets subject to *miR-490-3p* regulation in colorectal cancer (CRC) cells. To identify genes controlled by *miR-490-3p* in CRC cells, we used the TargetScan database and gene expression profile of *miR-490-3p* transfected HCT116 cells (GEO accession number: GSE129043). Moreover, GEPIA2 database analysis was used to identify upregulated genes in CRC tissues. A total of 38 genes were identified as possibly controlled by *miR-490-3p* in CRC cells.

**Figure 6 ijms-22-09876-f006:**
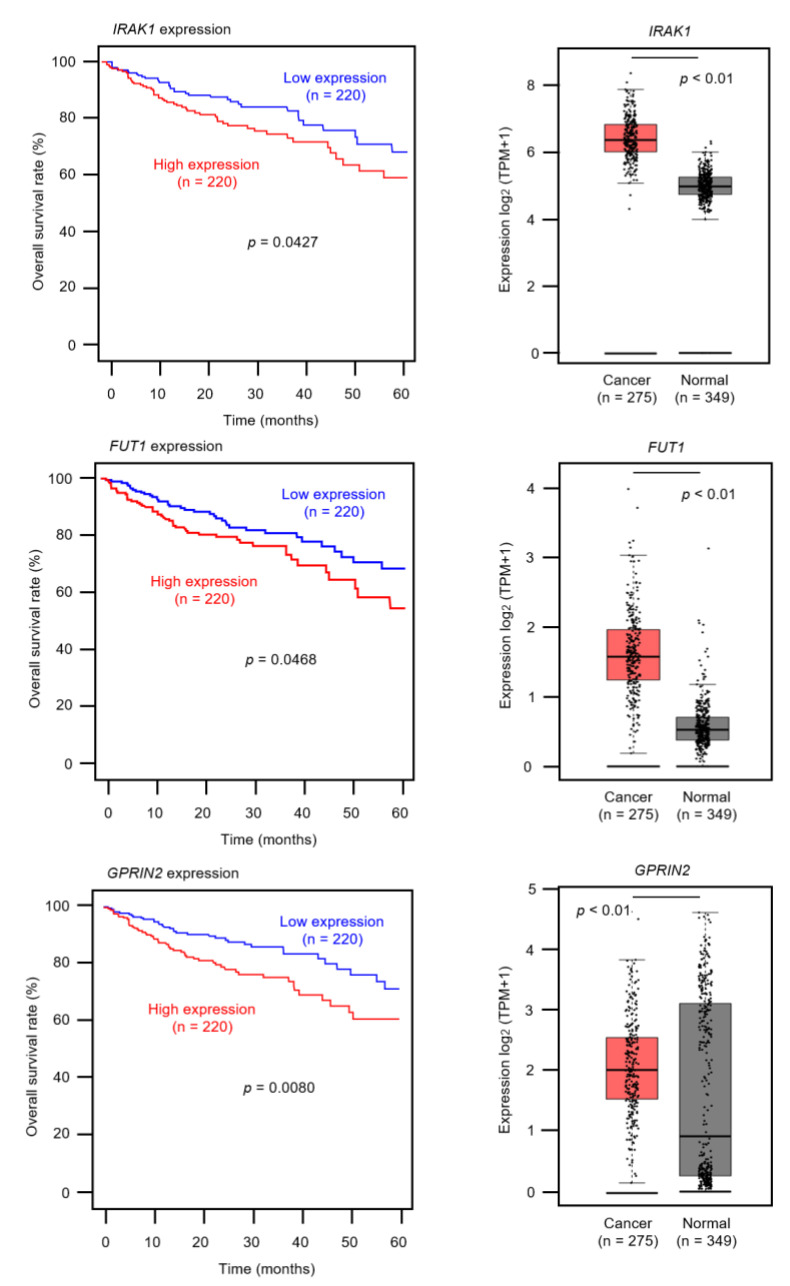
Clinical significance of *miR-490-3p* target genes (*IRAK1*, *FUT1,* and *GPRIN2*) in CRC cohort data analysis. Among 38 putative target genes, 3 genes (*IRAK1*, *FUT1* and *GPRIN2*) were significantly associated with poor prognosis in patients with colorectal cancer (*p* < 0.05). Kaplan–Meier curves for 5-year overall survivals of 3 genes are shown at left side. Patients were divided into high and low groups (relative to median expression) according to miRNA expression. The red line shows the high expression group, and the blue line shows the low expression group. Expression levels of three genes (*IRAK1*, *FUT1,* and *GPRIN2*) in CRC tissues and normal tissues obtained from TCGA-COAD, based on the GEPIA2 platform, are shown at the right side.

**Figure 7 ijms-22-09876-f007:**
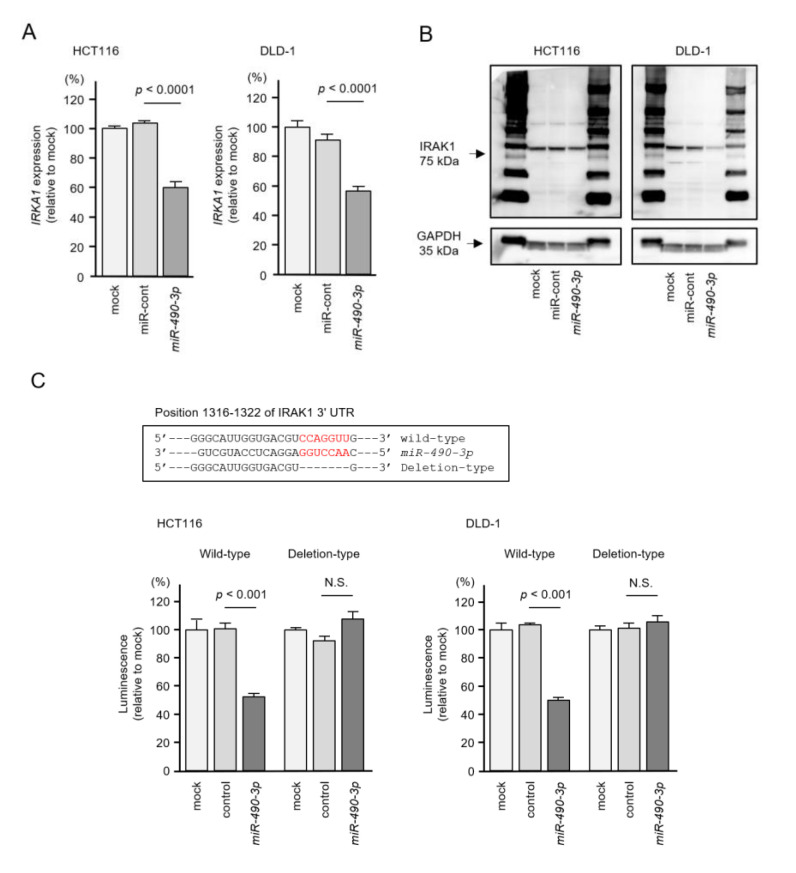
Direct regulation of *IRAK1* expression by *miR-490-3p* in colorectal cancer (CRC) cells (HCT116 and DLD-1). (**A**) Real-time PCR showing significantly reduced expression of *IRAK1* mRNA at 72 h after *miR-490-3p* transfection in CRC cells. Expression of *GAPDH* was used as an internal control. (**B**) Western blot showing reduced expression of IRAK1 protein at 72 h after *miR-490-3p* transfection in CRC cells. Expression of GAPDH was used as an internal control. (**C**) One putative *miR-490-3p* binding site predicted within the 3′-UTR of *IRAK1* by TargetScanHuman analyses (upper panel). Dual luciferase reporter assays showed reduced luminescence activity after cotransfection of the wild-type *IRAK1* 3′-UTR sequence (containing the *miR-490-3p* binding site) with *miR-490-3p* in CRC cells (lower panel). Normalized data were calculated as the Renilla/firefly luciferase activity ratio (N.S., not significant).

**Figure 8 ijms-22-09876-f008:**
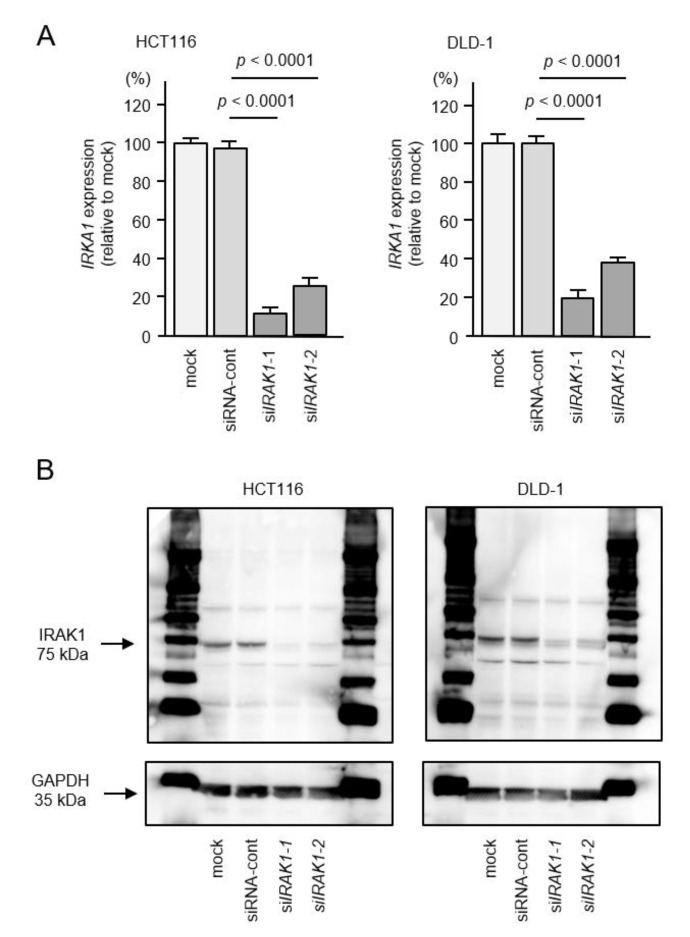
Knockdown efficiencies of *IRAK1* expression by siRNAs in colorectal cancer (CRC) cells. Two different siRNAs, *siIRAK1*-1 and *siIRAK1*-2 were used. (**A**) Real-time PCR showing significantly reduced expression of *IRAK1* mRNA 72 h after either s*iIRAK1*-1 or *siIRAK1*-2 transfection in CRC cells (HCT116 and DLD-1). Expression of *GAPDH* was used as an internal control. (**B**) Western blots showing reduced expression of *IRAK1* protein 72 h after *siIRAK1*-1 or *siIRAK1*-2 transfection in CRC cells. Expression of GAPDH was used as an internal control.

**Figure 9 ijms-22-09876-f009:**
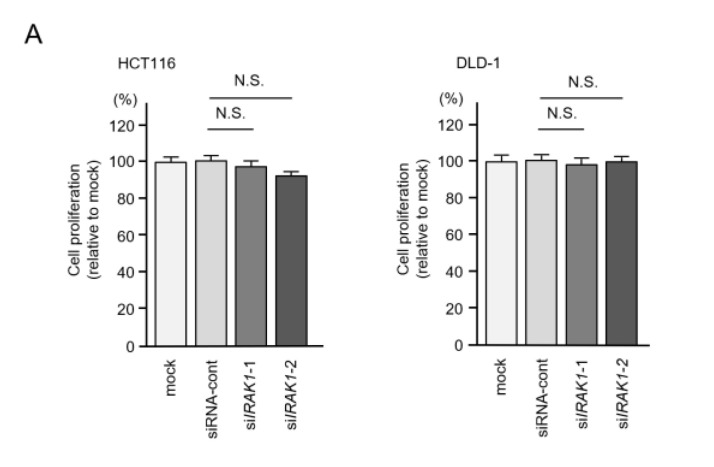
Effects of knockdown of *IRAK1* in colorectal cancer (CRC) cells (HCT116 and DLD-1). Functional assays of *IRAK1* (si*IRAK1*-1 and si*IRAK1*-2 transfection) in CRC cells. (**A**) Cell proliferation assessed by XTT assay 72 h after transfection of siRNA (**B**) Cell invasion determined by Matrigel invasion assay 48 h after seeding inhibitor-transfected cells into the chambers. (**C**) Cell migration assessed using a membrane culture system 48 h after seeding inhibitor-transfected cells into the chambers.

**Figure 10 ijms-22-09876-f010:**
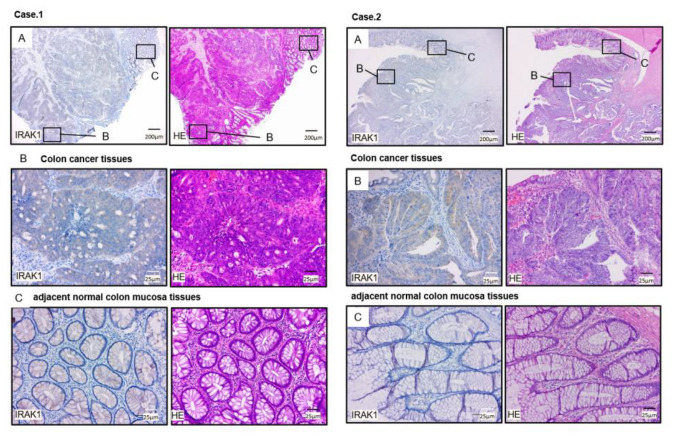
IRAK1 was high expression in colorectal tissues compared to adjacent normal colon tissues as demonstrated. The representative clinical sections of samples from case 1 (patient number 8) and case 2 (patient number 17) are shown. (**A**) Microscopic finding of immunohistochemical staining for IRAK1 on the left side and hematoxylin and eosin (H&E) staining on the right side were shown (magnification, ×20). Areas in the boxes are shown magnified at colon cancer tissue (**B**) and adjacent normal colon mucosa tissue (**C**). (**B**) Immunohistochemical staining for IRAK1 H&E staining in the colon cancer cell were shown (magnification, ×200). (**C**) Immunohistochemical staining for IRAK1 and H&E staining and in the normal colon epithelial cells were shown (magnification, ×200).

**Table 1 ijms-22-09876-t001:** Clinicopathological features of five colorectal cancer (CRC) patients whose CRC tissue and normal colorectal tissue were analyzed to generate the CRC miRNA signature.

No.	Age (Years)	Sex	Location	Differentiation	T	N	M	Stage	ly	v	Recurrence
1	66	Male	Rectum	Moderate	3	1a	0	IIIB	0	1	—
2	66	Male	Colon(S)	Moderate	3	1a	0	IIIB	1	1	—
3	79	Male	Rectum	Moderate	3	2a	0	IIIB	1	1	—
4	78	Female	Colon(S)	Moderate	3	0	0	IIA	0	1	—
5	83	Female	Colon(S)	Moderate	3	1a	0	IIIB	1	0	—

ly, lymphatic invasion; M, metastasis; N, nodes; T, tumor; v, venous invasion; colon (S), sigmoid colon.

**Table 2 ijms-22-09876-t002:** Candidate genes of *miR-490-3p* targets in CRC cells.

Entrez Gene ID	Gene Symbol	Gene Name	Binding Sites	HCT116 *miR-490-3p* Transfectant log_2_ FC ≤ 1	OncoLnc 5 yrs (*p-Value* *)
9721	*GPRIN2*	G protein regulated inducer of neurite outgrowth 2	1	−2.3130598	**0.008**
3654	*IRAK1*	interleukin-1 receptor-associated kinase 1	1	−1.0598125	**0.0427**
2523	*FUT1*	fucosyltransferase 1 (galactoside 2-alpha-L-fucosyltransferase, H blood group)	1	−1.5235968	**0.0468**
135112	*NCOA7*	nuclear receptor coactivator 7	1	−1.177208	0.0585
84061	*MAGT1*	magnesium transporter 1	1	−1.5050192	0.0799
6382	*SDC1*	syndecan 1	1	−1.0336791	0.088
3635	*INPP5D*	inositol polyphosphate-5-phosphatase, 145 kDa	1	−1.2042127	0.1059
3178	*HNRNPA1*	heterogeneous nuclear ribonucleoprotein A1	3	−1.6466646	0.1496
84152	*PPP1R1B*	protein phosphatase 1, regulatory (inhibitor) subunit 1B	1	−2.459818	0.163
56886	*UGGT1*	UDP-glucose glycoprotein glucosyltransferase 1	1	−1.0872145	0.2326
8529	*CYP4F2*	cytochrome P450, family 4, subfamily F, polypeptide 2	1	−2.5992675	0.2469
647024	*C6orf132*	chromosome 6 open reading frame 132	2	−1.2631998	0.3329
10525	*HYOU1*	hypoxia up-regulated 1	2	−1.6224588	0.3420
2444	*FRK*	fyn-related kinase	2	−1.2127504	0.3546
154796	*AMOT*	angiomotin	1	−1.0376037	0.3709
23446	*SLC44A1*	solute carrier family 44 (choline transporter), member 1	2	−1.1456499	0.4167
440145	*MZT1*	mitotic spindle organizing protein 1	1	−1.0596924	0.4636
28985	*MCTS1*	malignant T cell amplified sequence 1	1	−1.0754105	0.4927
4642	*MYO1D*	myosin ID	1	−1.1291242	0.5085
84187	*TMEM164*	transmembrane protein 164	1	−1.0723546	0.5172
80223	*RAB11FIP1*	RAB11 family interacting protein 1 (class I)	2	−1.374427	0.5477
8635	*RNASET2*	ribonuclease T2	1	−1.4826338	0.5618
3146	*HMGB1*	high mobility group box 1	1	−1.8401318	0.5719
340706	*VWA2*	von Willebrand factor A domain containing 2	1	−1.3371019	0.6107
51561	*IL23A*	interleukin 23, alpha subunit p19	1	−1.1564418	0.6474
6850	*SYK*	spleen tyrosine kinase	1	−1.065637	0.6535
2232	*FDXR*	ferredoxin reductase	1	−1.0080034	0.7008
2525	*FUT3*	fucosyltransferase 3 (galactoside 3(4)-L-fucosyltransferase, Lewis blood group)	1	−2.2499368	0.7117
7416	*VDAC1*	voltage-dependent anion channel 1	2	−1.3866558	0.7136
80201	*HKDC1*	hexokinase domain containing 1	1	−2.3432722	0.7136
196264	*MPZL3*	myelin protein zero-like 3	1	−1.0834669	0.7250
3703	*STT3A*	STT3A, subunit of the oligosaccharyltransferase complex (catalytic)	1	−1.6853191	0.7465
7039	*TGFA*	transforming growth factor, alpha	2	−1.0281087	0.7569
4494	*MT1F*	metallothionein 1F	1	−1.7403115	0.7756
6523	*SLC5A1*	solute carrier family 5 (sodium/glucose cotransporter), member 1	1	−2.3984282	0.8507
6653	*SORL1*	sortilin-related receptor, L (DLR class) A repeats containing	1	−1.0742446	0.853
201595	*STT3B*	STT3B, subunit of the oligosaccharyltransferase complex (catalytic)	1	−2.0069218	0.0046
345079	*SOWAHB*	sosondowah ankyrin repeat domain family member B	1	−1.8673139	0.0519

* Kaplan-Meier analysis log-rank *p*-value < 0.05. Bold values, Kaplan-Meier analysis log-rank *p*-value < 0.05, poor prognosis with a high expression. Italic value, Kaplan-Meier analysis log-rank *p*-value < 0.05, poor prognosis with a low expression.

## Data Availability

The data presented in this study are available on request from the corresponding author.

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
