# Peer review of "RNA-Sequencing Based microRNA Expression Signature of Colorectal Cancer: The Impact of Oncogenic Targets Regulated by *miR-490-3p"

_ijms, 2021, doi:10.3390/ijms22189876_

Round 1

Reviewer 1 Report

The manuscript profiles a small number of colorectal cancer patients, identifies Up- and Down-regulated miRNAs then looks in further details at one of these miRNAs (miR-490). It predicts miR-490 target genes and does a basic demonstration of the capacity of miR-490 to target the selected gene (IRAK1).

The study is well presented and results clearly displayed. Whilst there is little evidence to speak to the importance of miR-490 or its targeting of IRAK1 in CRC (there are MANY dysregulated miRNAs, miRNAs have MANY targets etc), it nevertheless clearly shows the results it reports given the studies limitations.

My strongly suggested points to consider are two fold:

1) The study is based upon the smRNA seq of only 10 patients (5 CRC, 5 controls). As is typical, a vast number of "dysregulated" miRNAs are reported. The small sample size has no way to control for the many variables that exist between people and so little can really been concluded from the tables. I would encourage a comparative analysis between the authors data and any larger publically available smRNA-Seq data the authors can find to see which miRNAs (including -490) overlap.

2) On a related note, it is common in such studies for a normal vs cancer difference to actually be reflecting something else. Especially given Fig 3 where you go from high miR-490 expression in normal tissue to absent for the cancer. How do you know this is not due to an infiltrate of a cell population. ie: if the cancer was epithelial, the normal sample included mesenchymal stroma and miR-490 is simply a mesenchymally expressed miRNA? The authors should at the very least discuss and acknowledge this possibility (or refute this if they feel it appropriate).

Reviewer 2 Report

In their publication ‚RNA-sequencing based microRNA expression signature of colorectal cancer: the impact of oncogenic targets regulated by miR-490-3p’ the authors performed next generation sequencing for small RNA’s on five patient samples and compared it to five samples of tumor adjacent tissues to identify a total of 154 differentially regulated miRNAs. They made the interesting observation that from 16 pre-mRNAs the deduced 5’ as well as 3’ miRNAs were differentially regulated. Transfecting miR-490-3p and miR-490-5p into two colon cancer cell lines both miRNAs reduced proliferation, invasion and migration. Instead of pursuing this comparison further, the authors decided to identify new targets of miR-490-3p. Subsequent to an in-silico approach, they demonstrated that target-site containing IRAK1, FUT1 and GPRIN2 had an impact on the overall survival of CRC patients – as deduced from the TCGA-COAD database. Dual luciferase reporter assays were applied next to confirm IRAK1 as a direct of miR-490-3p. Subsequently, siRNA-mediated knockdown of IRAK1 in CRC cell lines showed reduced invasion and migration, but not proliferation. Finally, immunohistochemical stainings of two tumor and two tumor-adjacent samples were presented to demonstrated increased levels of IRAK1 protein in tumor samples.

Overall, the manuscript was well prepared and the English is adequate. The experiments follow a standard procedure. Following a screen, the outcome was validated and targets were identified.

Here are my major concerns:

  1. Searching for ‘miRNA, CRC and cancer’ there are ~2600 publications in pubmed. ‘miR-490-3p and cancer’ delivers 75 results and ‘miR-490-3p and CRC’ comes back with four publications. At least three publication already identified miR-490-3p targets, including TGFbR1 (Xu, X., 2015), RAB14 (Wang, B., 2018) and VDAC1 (Liu, X., 2018). In my opinion, the authors neglected to integrate their work into the body of research that has previously been accumulated. For instance, I expect to find a paragraph in the introduction on the topic of ‘miRNA and CRC’. Also, it needs to be mentioned in prominent position in the introduction that miR-490-3p has been established as a differentially regulated miRNA in CRC prior to this work. I realize that this is a topic in the discussion, but researcher don’t always read the discussion of a publication and hence this point might be missed.
  2. The authors should provide a subfigure that compares miRNAs that were previously identified as differentially regulated in CRC to their own results.
  3. The methods were described rudimentary. Especially the major methods of the manuscript need to include key information, so that a skilled scientist can comprehend what was actually done. It is not good to have information scattered in tables and other publications. For instance, supplementary table 2 is quite small and the information can easily be integrated into the methods text. The kind of method that was used to arrive at a result is also part of every Figure legend. When the authors measured miRNA levels in Fig. 3, I went from the Figure legend to the methods to Table S2 to finally understand how they performed the experiment.
  4. I like the chromosome mapping of the differentially regulated miRNAs, but the tables should not be part of the main text. This is quite redundant and takes up a lot of space. The tables should be supplementary excel tables that include all the miRNAs that were measured - with the ones that were significantly regulated marked/highlighted. This way other researchers can find their favorite miRNAs even if they maybe just didn’t make the cut. I also think that the NGS sequencing data should be deposited at a public repository to provide colleagues with the option to perform meta analysis.
  5. The authors abandon the interesting miR-490-3p/miR-490-5p analysis too fast. During miRNA processing, the pre-miRNA is cleaved by dicer and the strand with the thermodynamically less stable 5’ end is transferred to the Ago protein. I am wondering if the 16 identified miRNA pairs might be similarly stable on both ends? Of the 16 miRNA pairs the 5’ miRNA should be measured by qPCR relative the 3’ miRNA. It will become clear if one of the two strands is predominantly expressed or if both are present at similar levels. There could be a scenario in which miR-490-3p and -5p are both are differentially expressed but still one of the two miRNAs is present at such a low level that it just doesn’t matter for the regulation of target genes. As pre-miRNAs are bound by different RNA-binding proteins (Treiber et al, Mol. Cell, 2017) it might also be possible that the identified 16 miRNA pairs (pre-miRNAs) are processed differently as compared to other pre-miRNAs. This aspect should be explored a little bit further.
  6. Figure 10 is supposed to demonstrate higher IRAK1 levels in tumor tissues. However, the architecture of the tumor and the tumor adjacent tissue are so different that I fail to see this. I don’t think that expression levels can be easily compared by IHC if the tissues are so different. Based on data presented in previous figures of this manuscript I am convinced that increased IRAK1 expression is likely to be a consequence of miR-490-3p regulation, but I don’t think that figure 10 can be presented as is without further reaching experiments or – at the very least – more detailed explanations.

Minor points:

  1. The authors clearly obtain less miRNA reads from cancer samples (2.7 fold). They may want to mention this, because it is in line with the finding that miRNAs are generally expressed at lower levels in cancer tissues (Lu J, Nature, 2005).
  2. The term ‘overexpression’ that was used in lines 241 and 244 is reserved for increased expression levels as a consequence of molecular manipulations and should not be used in this context.
  3. Figure 7b/8b: The writing should not be within the image.
  4. Line 212: … two different siRNAs… rather then … two types of siRNAs.
  5. Line 223: …no significant effects of these siRNA (not miRNAs – they don’t function in the same way!)
  6. The first paragraph of the discussion is not a classical discussion and on top is redundant with the introduction.
  7. Line 296 .. will reveal the biological characteristics .. change to …. will likely/possibly reveal the biological characteristics.
